# DrugReasoner: Interpretable drug approval prediction with a reasoning-augmented language model

Mohammadreza Ghaffarzadeh-Esfahani[1], Ali Motahharynia[1,2]*, Nahid Yousefian[1], Navid Mazrouei[1], Jafar Ghaisari[3], Yousof Gheisari[1]

1 Regenerative Medicine Research Center, Isfahan University of Medical Sciences, Isfahan, Iran,
2 Isfahan Neuroscience Research Center, Isfahan University of Medical Sciences, Isfahan, Iran,
3 Department of Electrical and Computer Engineering, Isfahan University of Technology, Isfahan, Iran

* alimotahharynia@gmail.com

## Abstract

Drug discovery is a complex and resource-intensive process, making early prediction of approval outcomes critical for optimizing research investments. While classical machine learning and deep learning methods have shown promise in drug approval prediction, their limited interpretability constraints their impact. Here, we present DrugReasoner, a reasoning-based large language model (LLM) built on the LLaMA architecture and fine-tuned with group relative policy optimization (GRPO) to predict the likelihood of small-molecule approval. DrugReasoner integrates molecular descriptors with comparative reasoning against structurally similar approved and unapproved compounds, generating predictions alongside step-by-step rationales and confidence scores. DrugReasoner achieved robust performance with an AUC of 0.732 and an F1 score of 0.729 on the validation set and 0.725 and 0.718 on the test set, respectively. These results outperformed conventional baselines, including logistic regression, support vector machine, and k-nearest neighbors and had competitive performance relative to XGBoost. On an external independent dataset, DrugReasoner outperformed both baseline and the recently developed ChemAP model, achieving an AUC of 0.728 and an F1-score of 0.774, while maintaining high precision and balanced sensitivity, demonstrating robustness in real-world scenarios. These findings demonstrate that DrugReasoner not only delivers competitive predictive accuracy but also enhances transparency through its reasoning outputs, thereby addressing a key bottleneck in AI-assisted drug discovery. This study highlights the potential of reasoning-augmented LLMs as interpretable and effective tools for pharmaceutical decision-making.

**Data availability statement:** All data generated or analyzed during this study are included in the manuscript and supporting information files. The drug approval prediction dataset used in this study is publicly available at: "Moreza009/drug_approval_prediction" on Hugging Face (DOI:10.57967/hf/6497).

**Code availability statement:** The checkpoints and code for assessment of drug approval are publicly available at https://github.com/mohammad-gh009/DrugReasoner and "Moreza009/Llama-DrugReasoner" on Hugging Face (DOI:10.57967/hf/6496).Explore the interactive user interface of DrugReasoner at https://www.kaggle.com/code/mohammadgh009/drugreasoner.

**Funding:** The author(s) received no specific funding for this work.

**Competing interests:** The authors have declared that no competing interests exist.

## Introduction

Drug discovery is a complex and costly process, often requiring over a decade and substantial financial investment, reaching approximately 879 million dollars, to bring a single compound from discovery to market [1,2]. This makes early screening and accurate prediction of a drug candidate's eventual success crucial for optimizing resource allocation. In recent years, computational methods, including classical machine learning and deep learning, have contributed to improve drug discovery processes from structural validation to toxicity and efficacy screening [3]. A notable example is ChemAP [4], which applies knowledge distillation from a teacher model that integrates multi-modal information into a student model capable of predicting drug approval from chemical structures alone. However, such models remain constrained by their limited interpretability, making it challenging to build trust for model-driven decisions [5].

Large language models (LLMs) have recently emerged as transformative tools in artificial intelligence (AI), demonstrating broad capabilities across a wide range of domains through training on massive and diverse datasets [6]. A central advancement that has enhanced their precision and problem-solving capacity is the incorporation of chain-of-thought (CoT) reasoning, which allows LLMs to simulate human-like reasoning and improve the interpretability of their outputs [7]. These reasoning abilities have attracted considerable interest in the field of drug discovery, where the complexity of biological systems and the need for integrative analysis present significant challenges [8]. Recent efforts, including frameworks such as DrugReAlign [9] and DrugAgent [10], have applied agentic AI approaches that extend LLMs with specialized tools for information retrieval, knowledge integration, and decision support. By combining CoT reasoning with structured access to domain-specific data, these systems enable novel applications such as drug repurposing. On the other hand, models like MolReasoner [11] and Mol-R1 [12] have focused on leveraging these reasoning capabilities to fine-tune models for *de novo* molecular design, enabling the generation of novel chemical structures with desirable pharmacological properties, highlighting a paradigm shift toward reasoning-augmented LLMs in drug discovery.

In this study, we introduce DrugReasoner, an interpretable LLM built on the LLaMA architecture [13], designed to predict the likelihood of a drug approval based on molecular features. The reasoning capability of DrugReasoner is central to this task, as approval decisions depend on integrating complex, multifaceted evidence into a coherent judgment. By fine-tuning the model with group relative policy optimization (GRPO) [14] on a curated dataset containing both approved and unapproved compounds, as well as structurally similar approved and unapproved molecules for each entry, DrugReasoner is able to not only produce accurate predictions but also to articulate its decision-making process. Comparative evaluation against baseline and recent approval predictor models demonstrates that DrugReasoner achieves both strong predictive performance and enhanced interpretability, underscoring the importance of reasoning-augmented LLMs for transparent and efficient AI-assisted drug discovery.

## Results

In order to address the challenge of drug approval prediction, we developed DrugReasoner, an algorithm based on Llama-3.1-8B-Instruct. By fine-tuning it using GRPO with customized reward functions, this framework is designed to generate CoT reasoning and accurately predict small-molecule approval.

The model first analyzes the molecular features of the query compound, then compares them with the molecular features of the five most similar approved and unapproved molecules from the training set. Following this, it outputs a binary label (approved/unapproved) along with an explanatory rationale (Fig 1).

### DrugReasoner is effectively trained to interpret small-molecule approval

DrugReasoner was fine-tuned on a dataset of 2,255 approved and 2,255 unapproved small molecules, using customized reward functions that encouraged accurate predictions and coherent reasoning. Training was performed over 14,500 optimization steps, during which the model generated four outputs per input. The reward diagram is presented in Fig 2A (S1 File). Performance was monitored every 500 steps on the evaluation set. Checkpoint 12,500 was selected as the final model based on the AUC and other evaluation metrics and complete adherence (100%) to the expected output structure (Fig 2B, S2 File). In addition, the model was required to produce a confidence score for each prediction. This score stabilized at 0.87 toward the end of training. An example of model input formatting and reasoning output is shown in S3 File.

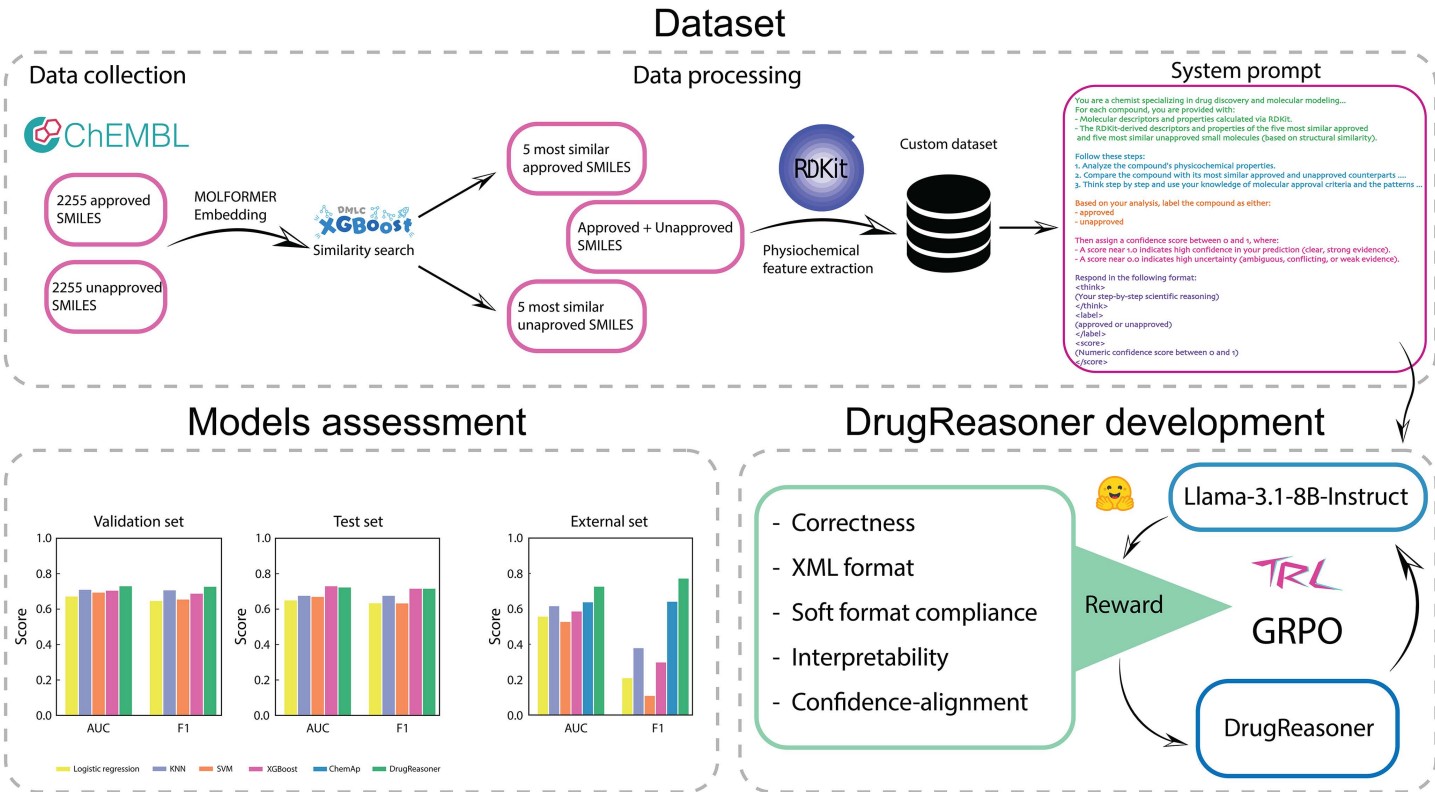

**Fig 1. Schematic representation of DrugReasoner development and assessment.** The upper panel illustrates dataset preparation and processing. The lower panel shows model training with group relative policy optimization (GRPO) using customized reward functions, followed by comparative evaluation against other models on validation, test, and external datasets.

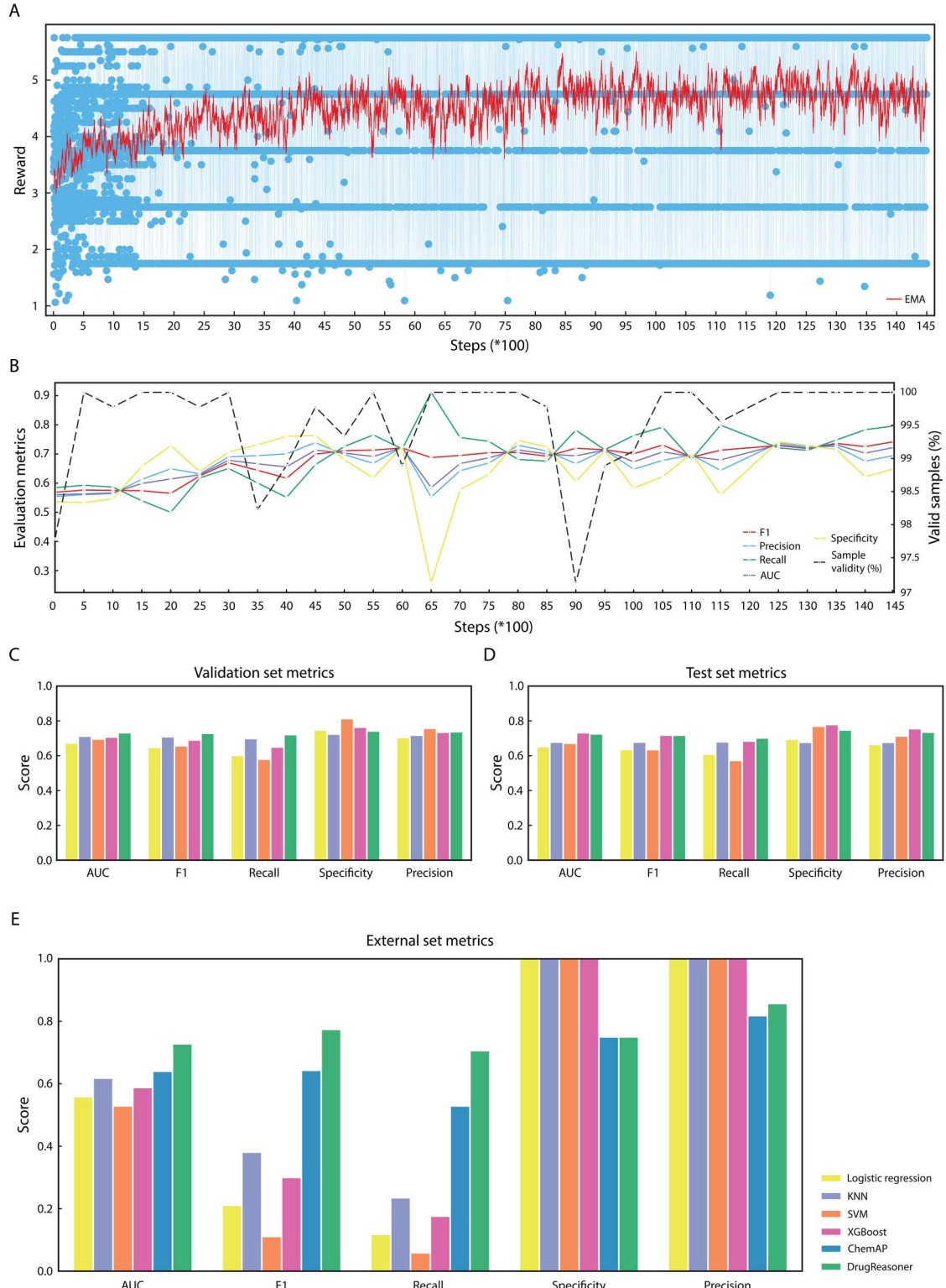

**Fig 2. DrugReasoner assessment. (A)** Reward trajectory during group relative policy optimization (GRPO) training with exponential moving average (EMA). **(B)** Evaluation metrics of model on the validation set. Comparative performance of DrugReasoner and baseline models on the **(C)** validation and **(D)** test sets. **(E)** Comparative performance of DrugReasoner with baseline models and ChemAP on an independent external dataset.

## DrugReasoner demonstrated reasonable performance in comparison to baseline models

DrugReasoner outperformed four baseline models, k-nearest neighbors (KNN), logistic regression, support vector machine (SVM), and XGBoost [15] along with Ghose filter and the quantitative estimate of drug-likeness (QED) on the validation set, achieving the highest overall balance of metrics (AUC = 0.732, accuracy = 0.732, recall = 0.721, precision = 0.738, specificity = 0.742, F1 = 0.729, Fig 2C, S4 and S5 Files). It maintained robust performance on the test set (AUC = 0.725, accuracy = 0.725) and achieved the highest recall (0.702). Furthermore, it matched XGBoost's F1-score (0.718) with competitive precision (0.735) and specificity (0.748) (Fig 2D, S4 and S5 Files).

## DrugReasoner outperformed baseline and recently developed approval prediction models on the external dataset

On the external independent dataset, containing 17 approved and 8 unapproved drugs (ChemAP study dataset), DrugReasoner outperformed all baseline and ChemAP models (Fig 2E, S4 and S5 Files). It achieved the highest AUC (0.728), F1-score (0.774), and precision (0.857), while maintaining a balanced accuracy (0.720) with strong recall (0.706) and specificity (0.750). In contrast, conventional machine learning baselines showed weaker discriminative ability (AUC = 0.529 to 0.618) and minimal sensitivity (≤ 0.235) despite perfect specificity. Also, Ghose filter (AUC = 0.49 and F1 = 0.55) and QED (AUC = 0.61 and F1 = 0.59) showed limited discriminative ability profiles (S4 and S5 File). Moreover, ChemAP exhibited lower AUC (0.64), recall (0.529), and specificity (0.75), indicating its limited generalizability, which highlights DrugReasoner's superior real-world applicability.

## Ablation analysis demonstrated the contribution of key components in DrugReasoner

To assess the impact of individual modules within the DrugReasoner pipeline, we performed targeted ablations. First, we evaluated the effect of removing the comparative reasoning module, which references similar approved and unapproved molecules. Without this module, model performance declined substantially. On the validation set, AUC dropped to 0.498 and F1-score to 0.050, while recall fell to 0.026 (S6 File). Similarly, on the test set, performance remained near random chance, with an AUC of 0.500 and an F1-score of 0.026. On the external dataset, although precision and specificity remained high, AUC decreased to 0.559 and F1-score to 0.210, reflecting a severe loss in discriminative power and sensitivity.

In the second ablation, we replaced the reasoning module with a direct prediction approach that used only MOLFORMER-extracted molecular features classified via XGBoost. This configuration yielded strong performance on the validation and test sets, with a validation AUC of 0.760 and F1-score of 0.759, and a test set AUC of 0.785 and F1-score of 0.790 (S6 File). However, on the external dataset, this version showed a marked drop in generalization, with AUC falling to 0.551 and F1-score to 0.480, indicating reduced robustness in real-world scenarios.

To assess the role of the XGBoost classifier–which supports retrieval of similar approved and unapproved small molecules–we conducted a final ablation replacing it with logistic regression. We employed a direct prediction approach that used only MOLFORMER-extracted molecular features classified via logistic regression. This led to a noticeable decline across all datasets: validation AUC fell to 0.698 and F1-score to 0.699; test AUC dropped to 0.745 and F1-score to 0.748; and on the external dataset, performance further deteriorated to an AUC of 0.397 and an F1-score of 0.385 (S6 File). Overall, these results underscore the critical importance of the reasoning component for interpretability and generalization, and demonstrate that while MOLFORMER embeddings paired with XGBoost provide a strong similarity-based predictor, they lack the robustness achieved by the full DrugReasoner framework.

## Discussion

In this work, we presented DrugReasoner, a Llama-3.1-8B-Instruct algorithm, fine-tuned using GRPO and customized reward functions to predict drug approval outcomes while providing interpretable reasoning. The model achieved superior performance on the validation set compared to baseline models (logistic regression, KNN, SVM, and XGBoost) and classical drug-likeness metrics (Ghose filter and QED). On the test set, it surpassed or showed competitive results compared to

these models. On an external dataset, however, DrugReasoner significantly outperformed both baselines and the recently developed model, ChemAP [4]. These results highlight DrugReasoner's enhanced predictive accuracy, generalizability, and promise as a decision-support model in the drug discovery process.

The integration of reasoning capabilities into LLMs has emerged as a game-changing tool in computational drug discovery, enabling more interpretable and context-aware predictions across molecular tasks. For instance, models like CoTox [16] apply CoT reasoning to predict molecular toxicity by incorporating biological pathways and gene ontology, while MolReasoner [11] enhances interpretability through synthetic CoT samples and reinforcement learning to link chemical structures with linguistic descriptions. Similarly, frameworks such as DrugPilot [17] and DrugAgent [10] employ multi-agent collaboration and reasoning to automate workflows. They integrate domain-specific tools, addressing challenges in data processing and tasks like drug-target interaction modeling. Mol-R1 [12] further advances explicit long-CoT reasoning for molecule generation via iterative adaptation and distillation strategies, improving performance in knowledge-intensive domains. In this context, DrugReasoner extends reasoning-augmented LLMs to drug approval prediction by applying GRPO-fine-tuned CoT reasoning using molecular feature comparisons with structurally similar approved and unapproved compounds to generate binary outcomes, confidence scores, and explanatory rationales. By doing so, our model bridges predictive performance with interpretability in a domain where transparent decision-making is crucial.

Traditional machine learning approaches for drug approval prediction like DrugApp [18] have relied on diverse features, including molecular, physicochemical, clinical trial, and patent data, often achieving robust performance via ensemble methods. However, features like patent or clinical trial data are typically available only post-trial, limiting their use in early discovery. ChemAP addresses this gap by predicting approval from chemical structures via knowledge distillation, enriching semantic knowledge to distinguish approved from unapproved drugs. Furthermore, DrugReasoner introduces an LLM-based paradigm that operates on molecular features (derived solely from structures) to enhance prediction with interpretable reasoning. This enables superior results on external datasets compared to ChemAP and conventional baselines. By emphasizing coherent rationales and confidence scoring, our model not only improves predictive accuracy but also enhances trustworthiness, offering a new paradigm for early-stage drug discovery.

Despite promising results, this study has limitations that should be addressed in future studies. To mitigate the risk of potential data leakage from LLMs pretrained on vast internet datasets, we excluded simplified molecular-input line-entry system (SMILES) representations, relying instead on molecular features for improved interpretability. While this reduced the risk of bias, it may also have constrained the model's ability to capture fine-grained structural information. Computational constraints further limited our exploration of larger models. Extending the context window, which could have allowed the inclusion of SMILES for similar approved and unapproved molecules, and systematic hyperparameter searches during GRPO fine-tuning could further enhance the reasoning performance. Future work should address these constraints by integrating structural data in a controlled manner, optimizing hyperparameters, scaling model size, and context length to improve reasoning and predictive accuracy.

Overall, DrugReasoner represents a significant advancement toward interpretable reasoning-augmented LLMs for drug approval prediction. By combining CoT reasoning with GRPO fine-tuning and molecular feature comparisons, it achieves strong performance and generates explanatory rationales across validation, test, and external datasets. Its ability to operate solely on molecular features while maintaining generalizability, highlights its potential to accelerate pharmaceutical decision-making. As such, DrugReasoner establishes a foundation for transparent and trustworthy AI tools, capable of guiding critical investment and research decisions from the earliest stages of drug discovery.

## Methods

### Dataset preparation

A dataset of small molecules was obtained from the ChEMBL database (version 35) [19] including approved molecules in Phase IV clinical trials and molecules in the preclinical phase. Child molecules were replaced by their parent compounds

to avoid duplication. A subset of 2,255 approved and 2,255 unapproved molecules was created by random undersampling of the unapproved class. The SMILES structure was retrieved, and strings were canonicalized using RDKit (version 2023.9.5) [20].

The dataset was partitioned into training, validation, and test subsets (8:1:1) using a stratified sampling strategy to maintain class distribution across all splits. The validation set was used for hyperparameter optimization, and the test set was reserved for evaluating the final model. Additionally, an independent external dataset used in the ChemAP paper was used to assess real-world performance. This dataset contained 20 approved and 8 unapproved drugs. Three approved drugs overlapping with the training, validation, or test sets were removed, and the remaining molecules were used for external evaluation.

### Data processing

**Molecular embedding.** SMILES representations were embedded using MOLFORMER [21], a pretrained transformer model trained on SMILES strings via masked language modeling. MOLFORMER incorporates linear attention with rotary positional embeddings to enhance scalability and contextual encoding. Each molecule was tokenized using its tokenizer and embeddings were computed in batches of 64 with attention masking, mapping each molecule into a 768-dimensional embedding space. Final molecular embeddings were derived by applying mean pooling over the last hidden states, weighted by the attention mask to account for variable sequence lengths and padding.

**XGBoost similarity search.** To enhance the dataset and improve model performance, for each molecule, we identified the five most similar approved and the five most similar unapproved molecules. Similarity was computed using XGBoost (version 3.0.2), trained on the MOLFORMER embeddings within the training set only. The model was trained as a binary classifier (approved vs. unapproved) on the training set using molecular embeddings from MOLFORMER as input features, with log-loss (binary cross-entropy) as the evaluation metric. Hyperparameter optimization was performed using Optuna (version 3.6.1) [22] on 0.1 of the training set with the tree-structured parzen estimator sampler and median pruner for early stopping. The search space included learning rate, maximum tree depth, L1 and L2 regularization, and subsampling ratios. A total of 1,000 trials were conducted, with early stopping of 20 rounds based on validation log-loss. The optimal hyperparameter configuration was used to train the final XGBoost model on the full training set.

The trained XGBoost model was used to generate leaf embeddings for each molecule by recording the index of the terminal leaf node within each decision tree for a given input. Pairwise hamming distances between leaf embeddings quantified topological similarity in the decision space. In the intra-training set, we implemented a self-comparison procedure, where for each molecule in the training set, the five most similar approved and unapproved compounds were identified based on their leaf proximity. For validation and test sets, a query-based similarity function was employed. Each molecule was embedded via MOLFORMER, and its leaf trajectory was compared against the training set to identify the top five most similar molecules of each class.

**Feature extraction.** To prevent data leakage–given that the base model was trained on extensive internet-scale data–we excluded SMILES strings, which could allow the model to recognize known molecules directly. Instead, we used only computed molecular features, which also improved interpretability. For all molecules, including query molecules and their neighbors, we computed physicochemical and structural descriptors using RDKit. These included molecular weight, LogP, topological polar surface area, hydrogen bond donors and acceptors, rotatable bonds, molecular refractivity, chiral centers, heavy atoms, ring counts, and formal charge. Additionally, structural alert checks were performed using pan assay interference compounds and Brenk filters to assess undesirable substructures.

**Prompt instruction.** Each input sample included:

- RDKit descriptors of the candidate molecule

- RDKit descriptors of the five most similar approved molecules

- RDKit descriptors of the five most similar unapproved molecules

These features were encoded in a structured prompt, where the system prompt defined a domain-specific instruction set to simulate expert chemical reasoning (S3 File).

The model output consisted of three fields in structured XML tags:

- <think>: Explanation and comparative reasoning

- <label>: Binary decision (approved or unapproved)

- <score>: Confidence score (0.0 to 1.0)

This schema ensured interpretability and allowed structured reward modeling.

## DrugReasoner development

We fine-tuned Llama-3.1-8B-Instruct [23] using GRPO trainer from hugging face transformer reinforcement learning (TRL) library (version 0.15.2) [24] to perform CoT reasoning on molecular features.

Training utilized the unsloth library (version 2025.3.19) [25] with:

- Maximum sequence length: 4,096 tokens (prompt+completion, 2,048 each)

- 4-bit quantization [26]

- Low-rank adaptation (LoRA) [27] with rank 32, applied to key attention projection layers (q_proj, k_proj, v_proj, o_proj, gate_proj, up_proj, down_proj) to reduce memory overhead.

**Reinforcement learning with GRPO.** The model was trained using GRPO trainer for 14,500 steps, generating four outputs per prompt during training, which were scored using multi-objective rewards. Training was performed with a learning rate of $5 \times 10^{-6}$, Adam optimizer parameters $\beta_1 = 0.9$ and $\beta_2 = 0.99$, a weight decay of 0.1, 100 warmup steps, and a cosine learning rate scheduler. All other variables set to the default of the library. Optimization employed the paged_adamw_8bit optimizer [28] with a per-device batch size of 4 and gradient clipping at a maximum norm of 0.1.

Group relative policy optimization works by generating multiple possible responses or a group ($K_j$) of possible responses ($actions = \{a_{j1}, a_{j2}, \cdots, a_{jK_j}\}$) from the current policy ($\pi_\theta$) for each input ($s_j$). Each response in the group is evaluated and assigned a reward score ($R_{jk}$) based on the defined reward function. The average reward across the group ($\bar{R}_j$) serves as a baseline (Eq. 1), and the advantage ($A_{jk}$) of each response is calculated as the difference between its reward and this group mean (Eq. 2).

$$\bar{R}_j = \frac{1}{K_j} \sum_{k=1}^{K_j} R_{jk}$$

(1)

$$A_{jk} = R_{jk} - \bar{R}_j$$

(2)

Using this advantage metric, the model updates its parameters to reinforce responses with above-average rewards while discouraging those with below-average performance. To ensure training stability, a clipped surrogate objective ($L_{jk}(\theta)$) is applied, where the clipping function ($clip(., 1-\in, 1+\in)$ prevents updates from deviating excessively from the reference policy. Here, $\pi_{\theta_{old}}$ is the policy before the update and $\in$ is a small hyperparameter (Eq. 3 and 4).

$$r_{jk}(\theta) = \frac{\pi_\theta(a_{jk}|s_j)}{\pi_{\theta_{old}}(a_{jk}|s_j)}$$

(3)

$$L_{jk}(\theta) = \min\left(r_{jk}(\theta)A_{jk}, \text{clip}\left(r_{jk}(\theta), 1-\in, 1+\in\right)A_{jk}\right) \tag{4}$$

This optimization is guided by a specialized loss function ($\mathcal{L}(\theta)$) that also incorporates a KL-divergence penalty ($D_{KL}$) to prevent abrupt policy changes and ensure stable learning (Eq. 5).

$$\mathcal{L}(\theta) = -\sum_{j}\sum_{k}L_{jk}(\theta) + \beta\sum_{j}D_{KL}\left(\pi_{\theta_{old}}(\cdot\mid s_j)\,\|\,\pi_\theta(\cdot\mid s_j)\right) \tag{5}$$

**Reward functions.** We implemented a multi-faceted reward function strategy composed of five distinct objectives to perform the scientific reasoning task.

- Correctness: Compares predicted label to ground truth (2.0 for correct, 0.0 otherwise). This function also logs intermediate outputs for qualitative analysis.

- XML format: Assesses whether the response follows the specified format (presence and singularity of <think>, <label>, and <score> tags). Each correctly placed start (<>) and end (</>) tag contributes 0.125, summing to a maximum of 0.75 (6*0.125).

- Soft format compliance: A regular expression-based reward function that returns 0.5 if the response approximately matches the expected format, meaning it includes the required structural elements (e.g., specific tags) in the correct order with allowances for minor whitespace variations, extra text, or small ordering changes, and 0.0 if key elements are missing or the structure is significantly incorrect.

- Interpretability: Assigns 0.5 if the extracted label is semantically valid (i.e., approved or unapproved).

- Confidence-alignment: Evaluates consistency between the predicted label and its confidence score:

  ○ Correct with high confidence ($\geq 0.7$): 2.0

  ○ Correct with moderate confidence (0.4 - < 0.7): 1.0

  ○ Correct with low confidence (< 0.4): 0.0

  ○ Incorrect with low confidence (< 0.4): 1.0

  ○ Incorrect with moderate confidence (0.4 - < 0.7): 0.5

  ○ Incorrect with high confidence ($\geq 0.7$): 0.0

Each reward function returned a scalar value, which was used to score the model's generated completions during GRPO training. Reward signals were recorded at each step for inspection and debugging (Fig 2A and S1 File).

**Checkpoint selection.** Training was performed for a total of 14,500 steps. To identify the optimal checkpoint for inference, we tracked the trajectory of reward scores throughout training. In addition, every 500 steps, we performed inference using top_p = 0.9, top_k = 9, and temperature = 1, evaluating the model on the validation set with metrics including, AUC, F1, precision, recall, specificity, and accuracy. The checkpoint at 12,500 steps achieved optimal performance and was selected as the final model (Fig 2B and S2 File).

## Model evaluation

DrugReasoner was evaluated against four conventional baselines: logistic regression, SVM, KNN, and XGBoost. To evaluate the drug-likeness of the generated molecules, we also employed the Ghose filter and QED from RDKit. The Ghose filter assesses molecular properties such as molecular weight, logP, and molar refractivity to determine if a compound

falls within ranges typical for drug-like molecules. QED provides a continuous score reflecting overall drug-likeness based on multiple physicochemical features. Molecules with a QED score above 0.5 were considered drug-like. To assess real-world applicability, we compared DrugReasoner with ChemAP using the external validation dataset described in the ChemAP study. All baseline models, Ghose filter and QED, along with ChemAP and DrugReasoner, were evaluated on this external dataset.

### Ablation analysis

To evaluate the contribution of core components within the DrugReasoner pipeline, we performed a systematic ablation analysis. First, we removed the comparative module, which provides the model with molecular features of the five most structurally similar approved and unapproved compounds from the XGBoost classifier. This modification was evaluated to isolate the impact of explicit comparative reasoning on performance. Second, we ablated the entire reasoning framework and used the rest of the pipeline, where molecular embeddings extracted by MOLFORMER were used as input to a standalone XGBoost classifier. This setup assessed the baseline predictive power of the molecular features combined with a non-interpretable model. Third, to specifically evaluate the role of the XGBoost classifier within this non-reasoning pathway, we replaced it with logistic regression and measured its performance in discriminating approved from unapproved molecules.

### Hardware and training time

Training was performed on a single NVIDIA V100 GPU (32 GB VRAM) with an AMD CPU (64 GB RAM, 4 cores). Total training time was approximately 794.3 hours. Baseline models were evaluated on CPU-only hardware.

## Supporting information

**S1 File. Training performance data.** Complete record of reward values during DrugReasoner training.
(JSON)

**S2 File. Model checkpoint evaluations.** Detailed metrics tracked at various stages of training.
(CSV)

**S3 File. Prompt framework.** System instructions and an example user prompt provided to DrugReasoner.
(DOCX)

**S4 File. Comparative models evaluation.** Performance comparison between baseline models, Ghose filter, QED, ChemAP, and DrugReasoner across validation, test, and external validation datasets.
(DOCX)

**S5 File. Model prediction outputs.** Raw predictions generated by baseline models, Ghose filter, QED, ChemAP, and DrugReasoner on all evaluation datasets.
(XLSX)

**S6 File. Ablation study results.** Performance metrics during ablation analysis.
(DOCX)

## Author contributions

**Conceptualization:** Mohammadreza Ghaffarzadeh-Esfahani, Ali Motahharynia, Yousof Gheisari.

**Data curation:** Mohammadreza Ghaffarzadeh-Esfahani, Ali Motahharynia, Nahid Yousefian.

**Formal analysis:** Mohammadreza Ghaffarzadeh-Esfahani, Ali Motahharynia.

**Investigation:** Mohammadreza Ghaffarzadeh-Esfahani, Ali Motahharynia.

**Methodology:** Mohammadreza Ghaffarzadeh-Esfahani, Ali Motahharynia, Nahid Yousefian, Jafar Ghaisari, Yousof Gheisari.

**Software:** Mohammadreza Ghaffarzadeh-Esfahani, Ali Motahharynia, Nahid Yousefian.

**Supervision:** Ali Motahharynia.

**Validation:** Mohammadreza Ghaffarzadeh-Esfahani, Ali Motahharynia, Nahid Yousefian, Navid Mazrouei, Jafar Ghaisari, Yousof Gheisari.

**Visualization:** Mohammadreza Ghaffarzadeh-Esfahani, Ali Motahharynia.

**Writing – original draft:** Mohammadreza Ghaffarzadeh-Esfahani, Ali Motahharynia, Nahid Yousefian, Navid Mazrouei.

**Writing – review & editing:** Ali Motahharynia, Jafar Ghaisari, Yousof Gheisari.

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
