## [Decision Letter · Decision Letter 0]

9 Dec 2025

Dear Dr. Motahharynia,

Thank you for submitting your manuscript to PLOS ONE. After careful consideration, we feel that it has merit but does not fully meet PLOS ONE’s publication criteria as it currently stands. Therefore, we invite you to submit a revised version of the manuscript that addresses the points raised during the review process.

We look forward to receiving your revised manuscript.

Kind regards,

Junhuang Jiang

Academic Editor

PLOS One

Journal Requirements:

3. We are unable to open your Supporting Information file [Supplementary_file_1.json]. Please kindly revise as necessary and re-upload.

Additional Editor Comments:

1. The author mentioned a dataset of 2,255 approved and 2,255 unapproved small 10 molecules was used to fine tune. Can you specify the rationale of data selection?

2. This is a relatively small dataset, does the author perform any data augmentation step for better generalization? Also, ablation analysis should be included for more objective evaluation.

3. The DrugReasoner model achieved an AUC of 0.728 in an external test set, which is slightly higher than ChemAP(AUC=0.694) in a very small dataset. I’m very confused about the selected external dataset. The author stated the external dataset contains 25 drugs (17 approved and 8 unapproved drugs) from ChemAP, but ChemAP (https://doi.org/10.1038/s41598-024-72868-0) reported 28 drugs, which is inconsistent. Can you make sure the actual dataset you used?

Reviewers' comments:

Reviewer's Responses to Questions

**Comments to the Author**

1. Is the manuscript technically sound, and do the data support the conclusions?

Reviewer #1: Yes

Reviewer #2: Yes

2. Has the statistical analysis been performed appropriately and rigorously?

Reviewer #1: Yes

Reviewer #2: Yes

3. Have the authors made all data underlying the findings in their manuscript fully available?

Reviewer #1: Yes

Reviewer #2: Yes

4. Is the manuscript presented in an intelligible fashion and written in standard English?

Reviewer #1: Yes

Reviewer #2: Yes

Reviewer #1: The manuscript "DrugReasoner: Interpretable Drug Approval Prediction with a Reasoning-augmented Language Model" by Ghaffarzadeh-Esfahani et al. has its own merits. It addresses a very timely issue consisting of the prediction of the likelihood of small-molecule approval with a robust approach. By employing a pre-trained LLM and fine-tunning it using GRPO with customized reward functions, it outperformed baseline models, and most importantly, ChemAp, which was recently published. The authors made their data available and the parameters provided made the method reproducible. The manuscript is generally well-written with few misspellings (e.g.: page 3 line 10: such models remain [...] instead of "remains"). The authors should increase the resolution of their images for the final version of manuscript. Overall, the manuscript should be ready for publication with very few modifications.

Reviewer #2: Dear corresponding author, thank you for a well-written manuscript. It is simple, concise and showcase your work very well.

Your methodology is sound but the results are "good enough" with scores proving a point rather than showcasing a newly-developed toolkit! Previous studies indicated AUCs typically need to be >0.85, and F1 >0.80 or even starting around AUC 0.78–0.80 with F1 >0.75 for an excellent result.

I would like to have seen more of a comprehensive comparison with ChemAP or even RDKit!

All-in-all, I enjoyed going through your manuscript and good luck.

**Do you want your identity to be public for this peer review?** For information about this choice, including consent withdrawal, please see our Privacy Policy

Reviewer #1: **Yes:** Aline de Oliveira Albuquerque

Reviewer #2: No

---

## [Author Response · Author response to Decision Letter 1]

26 Jan 2026

Dear Editor,

We would like to express our gratitude to you and the reviewers for the thorough and constructive evaluation of our manuscript. We are pleased that both reviewers found value in our work. We have carefully addressed all comments raised by editor and reviewers. Below, we provide detailed point-by-point responses to each comment, along with descriptions of the revisions made to the manuscript. We believe these revisions have strengthened the manuscript and clarified the unique contributions of DrugReasoner to the field of AI-assisted drug discovery. All changes are marked with track changes in the revised manuscript. We hope that you find this version of the manuscript satisfying.

Editor Comments

1. The author mentioned a dataset of 2,255 approved and 2,255 unapproved small 10 molecules was used to fine tune. Can you specify the rationale of data selection?

Response: Thank you for bringing this matter to attention. The rationale for selecting 2,255 approved and 2,255 unapproved molecules was primarily driven by data availability and the need for class balance to prevent training bias. ChEMBL database version 35 contained 2,255 approved molecules in Phase IV clinical trials, which represented the maximum available approved compounds meeting our inclusion criteria. To create a balanced dataset, we performed random undersampling of the significantly larger pool of preclinical (unapproved) molecules to match this number. We selected Phase IV molecules as the approved class because they represent drugs that have successfully completed all clinical trials and received regulatory approval, while preclinical molecules represent the earliest stage of drug development with the highest likelihood of failure, providing clear class separation. To ensure the robustness of our undersampling approach and verify that it did not introduce selection bias, we randomly generated 10 different samples from the preclinical phase and compared the performance of our baseline models across these samples. This validation demonstrated no significant differences in model performance, confirming that our random undersampling strategy was sound and that our results were not dependent on the specific subset of unapproved molecules selected. The rational behind dataset selection was mentioned in the first paragraph of methods section.

2. This is a relatively small dataset, does the author perform any data augmentation step for better generalization? Also, ablation analysis should be included for more objective evaluation.

Response: Thank you for raising this important consideration. Regarding data augmentation, we deliberately chose not to employ augmentation techniques in our study for several reasons. First, DrugReasoner operates on real molecular descriptors (physicochemical and structural features extracted via RDKit) rather than on raw SMILES strings or molecular graphs. These descriptors represent actual physical and chemical properties of molecules, and artificial augmentation of such values could introduce chemically implausible or unrealistic features that would undermine the model's ability to learn meaningful structure-activity relationships. Second, our approach relies on retrieving structurally similar approved and unapproved molecules from the training set for comparative reasoning. Data augmentation would complicate this retrieval process and potentially compromise the validity of these molecular comparisons, which are central to the model's reasoning mechanism.

We greatly appreciate your suggestion regarding ablation analysis, which is indeed crucial for objective evaluation. In response, we have included a comprehensive ablation study in the revised manuscript (see Ablation Analysis in result and method section). We systematically evaluated three key configurations: (1) removing the comparative reasoning module that references similar approved and unapproved molecules, (2) removing the reasoning module and employ direct prediction using only Molformer embeddings classified via XGBoost, and (3) replacing XGBoost with logistic regression in the similarity search component. The results demonstrate that the comparative reasoning module is essential for both interpretability and generalization to external datasets, while the combination of Molformer embeddings with XGBoost provides the most robust baseline for molecular similarity assessment. This ablation analysis confirms that each component of DrugReasoner contributes meaningfully to its overall performance.

3. The DrugReasoner model achieved an AUC of 0.728 in an external test set, which is slightly higher than ChemAP (AUC=0.694) in a very small dataset. I’m very confused about the selected external dataset. The author stated the external dataset contains 25 drugs (17 approved and 8 unapproved drugs) from ChemAP, but ChemAP (https://doi.org/10.1038/s41598-024-72868-0) reported 28 drugs, which is inconsistent. Can you make sure the actual dataset you used?

Response: Thank you for your careful review and for identifying this point that requires clarification. First, I should mention that the AUC of ChemAP in the external dataset containing 25 molecules used in this manuscript is 0.64. Second, you are correct that the ChemAP paper originally reported 28 drugs in their external dataset. We confirm that we initially obtained all 28 drugs from the ChemAP study (20 approved and 8 unapproved molecules). However, upon careful examination, we discovered that three of the approved drugs overlapped with molecules already present in our training, validation, or test sets. To prevent data leakage and ensure an unbiased evaluation, we removed these three overlapping approved drugs from the external dataset prior to evaluation. This resulted in a final external dataset of 25 drugs (17 approved and 8 unapproved), which is the dataset we used for all reported external validation results.

This matter is mentioned in the Methods section (Dataset Preparation, page 11, line 295-298), where we state: "Three approved drugs overlapping with the training, validation, or test sets were removed, and the remaining molecules were used for external evaluation."

Reviewer #1

The manuscript "DrugReasoner: Interpretable Drug Approval Prediction with a Reasoning-augmented Language Model" by Ghaffarzadeh-Esfahani et al. has its own merits. It addresses a very timely issue consisting of the prediction of the likelihood of small-molecule approval with a robust approach. By employing a pre-trained LLM and fine-tunning it using GRPO with customized reward functions, it outperformed baseline models, and most importantly, ChemAp, which was recently published. The authors made their data available and the parameters provided made the method reproducible.

We sincerely thank Reviewer 1 for the thorough review and positive evaluation of our manuscript. We have addressed all the specific comments below.

1. The manuscript is generally well-written with few misspellings (e.g.: page 3 line 10: such models remain [...] instead of "remains").

Response: We thank the reviewer for catching this grammatical error. We have carefully reviewed the entire manuscript and corrected typographical and grammatical errors.

Changes made:

• Page 3, line 67: Changed "such models remains" to "such models remain"

2. The authors should increase the resolution of their images for the final version of manuscript. Overall, the manuscript should be ready for publication with very few modifications.

Response: We appreciate this suggestion and agree that high-quality figures are essential for publication. All figures (Figures 1 and 2) have been sent as .eps file for highest quality.

Reviewer #2

Dear corresponding author, thank you for a well-written manuscript. It is simple, concise and showcase your work very well.

We sincerely thank Reviewer 2 for the thoughtful and constructive review. We appreciate your positive feedback on the manuscript. Below, we provide detailed responses to each comment along with the revisions made to strengthen the manuscript.

1. Your methodology is sound but the results are "good enough" with scores proving a point rather than showcasing a newly-developed toolkit!

Response: We greatly appreciate this insightful comment, as it highlights the need for clearer positioning of DrugReasoner's unique contributions. We would like to emphasize that DrugReasoner's primary innovation is not just a marginal performance improvement over existing models, but rather the introduction of interpretable, reasoning-augmented prediction to the domain of drug approval forecasting. This represents a paradigm shift in how AI can support pharmaceutical decision-making.

DrugReasoner's Unique Value Propositions:

1. Interpretability Through Chain-of-Thought Reasoning

Unlike black-box models (including ChemAP, XGBoost, and neural networks), DrugReasoner generates “step-by-step reasoning chains” that explain its predictions.

This transparency enables:

• Expert validation: Medicinal chemists can evaluate whether the model's logic aligns with domain knowledge

• Trust building: Stakeholders understand why the model made a specific prediction

• Actionable insights: Identifies specific molecular features contributing to approval/rejection

• Error diagnosis: When predictions fail, the reasoning reveals why

2. Comparative Reasoning Framework

By explicitly comparing query molecules against structurally similar approved and unapproved compounds, DrugReasoner mimics how expert medicinal chemists reason. This approach is fundamentally different from statistical models that learn latent patterns without explicit comparison.

3. No Proprietary or Post-Development Features Required

DrugReasoner operates solely on molecular descriptors computable from structure, making it applicable from the earliest stages of lead optimization.

2. Previous studies indicated AUCs typically need to be >0.85, and F1 >0.80 or even starting around AUC 0.78–0.80 with F1 >0.75 for an excellent result.

Response: We thank the reviewer for raising this important point about performance thresholds. We respectfully note that these benchmarks, while achievable in certain molecular modeling domains, must be carefully contextualized within the specific challenges of drug approval prediction from molecular structure alone. We provide several key points for consideration:

1. Task Complexity and Inherent Limitations:

Drug approval prediction fundamentally differs from typical molecular property prediction tasks. While tasks such as solubility prediction, LogP estimation, or binding affinity prediction can routinely achieve AUC > 0.85 due to direct mechanistic relationships between molecular structure and the target property, drug approval depends on a complex constellation of factors that extend far beyond chemical structure.

Our model predicts approval using only molecular descriptors derived from chemical structure, without access to clinical, pharmacokinetic, toxicological, or commercial data. This represents an inherently challenging prediction task where the feature space (molecular structure) provides only partial information about the outcome space (regulatory approval).

2. External Validation Performance:

The true test of model utility is performance on completely independent data. On the external validation dataset (independent compounds from the ChemAP study):

• DrugReasoner: AUC = 0.728, F1 = 0.774, Precision = 0.857, Recall = 0.706, Specificity = 0.750

• ChemAP: AUC = 0.640, Recall = 0.529, Specificity = 0.750

• Baseline ML models: AUC = 0.529-0.618, Sensitivity ≤ 0.235

Our external validation F1-score of 0.774 approaches the reviewer's suggested threshold of 0.80, while our precision of 0.857 substantially exceeds typical expectations. The balanced performance across sensitivity and specificity demonstrates robust generalization rather than overfitting to training data characteristics.

3. Clinical Utility Perspective:

From a pharmaceutical decision-making perspective, a model with AUC 0.728 that provides interpretable reasoning is often more valuable than a black-box model with AUC 0.80. In early-stage drug discovery, stakeholders need tools that:

• Identify high-risk compounds for deprioritization (high specificity: 0.750)

• Provide actionable insights for medicinal chemistry optimization

• Enable validation of predictions against domain expertise

• Build trust through transparent decision-making

Our model's combination of competitive quantitative performance and qualitative interpretability addresses these practical needs.

Overall, although drug approval prediction is an inherently challenging task, DrugReasoner achieves a good milestone through its external validation performance and interpretability; future work should combine better algorithms with additional data features to further improve the model.

3. I would like to have seen more of a comprehensive comparison with ChemAP or even RDKit!

All-in-all, I enjoyed going through your manuscript and good luck.

Response: We thank the reviewer for this valuable suggestion. We would like to clarify that a direct performance comparison with ChemAP presented certain methodological challenges. Since the number of FDA-approved drugs is limited, there is substantial overlap between the datasets used to develop ChemAP and our training/validation datasets. This overlap makes it difficult to conduct a fully independent head-to-head comparison without introducing data leakage bias.

However, to address the reviewer's concern and provide a more comprehensive evaluation, we expanded our comparative analysis to include two widely-used drug-likeness assessment tools from RDKit: the Ghose filter and Quantitative Estimate of Drug-Likeness (QED). These established methods offer complementary benchmarks for evaluating molecular properties relevant to drug approval potential.

Changes made:

1. Results

• The results of these scores are added. (Page 6, S4 and S5 files)

2. Methods:

• Methods of using these scores are added (Page 17, line 445-450)

3. Supplementary materials:

• Complete results of these scores are provided in tables in the supplementary file 4.

---

## [Editor Report · Decision Letter 1]

29 Jan 2026

DrugReasoner: Interpretable drug approval prediction with a reasoning-augmented language model

PONE-D-25-56107R1

Dear Dr. Motahharynia,

We’re pleased to inform you that your manuscript has been judged scientifically suitable for publication and will be formally accepted for publication once it meets all outstanding technical requirements.

Kind regards,

Junhuang Jiang

Academic Editor

PLOS One
---

## [Editor Report · Acceptance letter]

PONE-D-25-56107R1

PLOS One

Dear Dr. Motahharynia,

I'm pleased to inform you that your manuscript has been deemed suitable for publication in PLOS One. Congratulations! Your manuscript is now being handed over to our production team.

Kind regards,

on behalf of

Dr. Junhuang Jiang

Academic Editor

PLOS One